# Estimating the equity impacts of the smoking ban in England on cotinine levels: a regression discontinuity design

Matthew Robson [iD],[1] Joseph Lord,[2] Tim Doran[1]

[1]Department of Health Sciences, University of York, York, UK
[2]Centre for Health Economics, University of York, York, UK

**Correspondence to**
Professor Tim Doran;
tim.doran@york.ac.uk

## ABSTRACT

**Objective** To estimate the equity impacts of the 2007 smoking ban in England, for both smokers and non-smokers.

**Design** Doubly robust regression discontinuity analysis of salivary cotinine levels. Conditional average treatment effects were used to estimate differential impacts of the ban by socioeconomic deprivation (based on the Index of Multiple Deprivation). Distributional impacts were further assessed using conditional quantile treatment effects and inequality treatment effects.

**Setting** In 2007, England introduced a ban on smoking in public places. This had little impact on tobacco consumption by smokers but was associated with decreases in environmental tobacco smoke exposure for non-smokers. However, the impact of the ban on socioeconomic inequalities in exposure is unclear.

**Participants** 766 smokers and 2952 non-smokers responding to the Health Survey for England in 2007.

**Outcome measure** Levels of salivary cotinine.

**Results** Before the ban, socioeconomic deprivation was associated with higher cotinine levels for non-smokers but not for smokers. The ban caused a significant reduction in average cotinine levels for non-smokers (p=0.043) but had no effect for smokers (p=0.817). Reductions for non-smokers were greater for more deprived groups with higher levels of exposure, and there was a significant reduction in socioeconomic-related inequality in cotinine. Across the whole population (both smokers and non-smokers), there was no significant increase in the concentration of cotinine levels among the socioeconomically deprived.

**Conclusion** The 2007 ban on smoking in public places had little impact on smokers, but was, as intended, associated with reductions in both (1) average levels of environmental tobacco smoke exposure and (2) deprivation-related inequality in exposure among non-smokers.

## STRENGTHS AND LIMITATIONS OF THIS STUDY

⇒ We used a doubly robust regression discontinuity design to derive causal estimates of the equity impacts of the smoking ban for both smokers and non-smokers.

⇒ Data were derived from a rich, individual-level data set with an objective measure of tobacco smoke exposure to avoid social desirability bias in reported variables.

⇒ Causal effects of the ban were estimated across the entire distribution of the outcome, identifying conditional distributional effects and the impact on socioeconomic-related inequality.

⇒ We focused only on the short-term effects of the ban and rely on an area-based measure of socioeconomic status.

## INTRODUCTION

Successive UK governments have pursued a range of public health policies to reduce population harms from tobacco smoke exposure since links to lung cancer, and other common fatal conditions were confirmed in the 1950s.[1] With public education, increased taxation and restrictions on sales and advertising, smoking prevalence fell from 70% in the 1950s to under 25% by the turn of the century.[2] More coercive policies began to be considered in the 1990s as studies emerged identifying harmful impacts on non-smokers exposed to environmental smoke,[3–12] including exposure at home[7 8] and in the workplace.[13] Estimates of the population impact of environmental exposure suggest that over 10% of smoking-related deaths may occur in non-smokers,[14] and the risks to smokers themselves are further increased by environmental smoke. The impacts of smoking are also unevenly distributed; while rates of smoking have fallen in all socioeconomic groups since 1950s, they fell fastest in the most affluent groups. At the end of the century, over half of the difference in mortality between men in high (professional, managerial and technical) and low (unskilled manual) socioeconomic groups was attributable to tobacco exposure.[15]

The growing weight of evidence on environmental tobacco smoke eventually led to the implementation of national bans in Scotland (March 2006), Wales and Northern Ireland (April 2007) and England (July 2007). For England and Wales, the Health Act 2006 effectively prohibited tobacco smoking in enclosed

public places, including workplaces, public transport and private clubs. The ban was found to have little impact on tobacco consumption in the short run,[16–20] but smokers were found to smoke substantially less in enclosed public places, including at work and in pubs.[17]

Evidence on the equity impacts of the smoking ban, particularly for non-smokers, is less clear. In England, smoking prevalence continued to fall at similar rates across different socioeconomic status (SES) groups after the ban, preserving the gaps between them; by 2015 smoking, prevalence remained above 25% in the most deprived fourth and fifth of areas, compared with 10% in the least deprived fifth.[19] In terms of secondhand exposure, Semple *et al*[21] noted that pubs in more deprived areas, with higher baseline particulate (PM2.5) concentrations, tended to have greater reductions after the ban. In contrast, King *et al*[22] found that higher SES smokers were more likely than low status smokers to prohibit smoking in their own home following the ban.

Obtaining more direct evidence on the equity impacts of smoking bans on non-smokers is subject to two key limitations: a reliance on before-and-after study designs measuring association, rather than strong econometric designs identifying causal effects and a focus on average effects for the whole population, rather than distributional effects relating to baseline exposure to tobacco smoke and socioeconomic circumstances. The latter limitation is particularly important, as national policies of this kind are frequently less effective for less affluent socioeconomic groups with higher baseline levels of exposure, and this has the unintended consequence of widening existing health inequalities. In this paper, we assess the impact of the national smoking ban on cotinine levels, a biomarker of tobacco smoke exposure, for both smokers and non-smokers. We use approaches that assess the full range of effects in relation to socioeconomic characteristics across the entire distribution of exposure levels, to more accurately determine the equity impacts of the smoking ban.

## METHODS
### Intervention
The national smoking ban in England was implemented on 1 July 2007. Smoking in enclosed public places became an offence, subject to fines of up to £200 for individuals and £2500 for businesses. Exemptions were included, at that time, for psychiatric units, nursing homes, prisons and other specified public places.

### Data
Data were derived from the Health Survey for England (HSE), an annual cross-sectional survey, which records health and lifestyle changes over time in England at the individual and household level. The data set includes variables covering tobacco smoking and exposure as well as a range of individual characteristics recorded by an interview and, if consent is given, there is a follow-up nurse's

visit (on average 18.4 days later). Our main analysis uses data from the 2007 wave of the survey, from January 2007 to January 2008, covering the period of the introduction of the public smoking ban in England.

### Patient and public involvement
As we conduct a secondary analysis of survey data, there was no direct patient and public involvement.

### Tobacco exposure
The primary outcome is exposure to tobacco smoke, estimated by salivary cotinine concentration. Cotinine is a metabolite of nicotine, which diffuses from the bloodstream into saliva, making salivary cotinine concentration a convenient and reliable biomarker of regular tobacco exposure.[23] Salivary cotinine concentrations above 12 ng/mL indicate smoking status with high sensitivity and specificity[24] and scale with the extent of tobacco use. Lower concentrations are also detectable, so cotinine is useful for monitoring changes in both tobacco use by smokers and exposure to secondhand smoke in non-smokers. Distributions of salivary cotinine levels tend to be heavily skewed; for non-smokers, levels are often undetectable, whereas for smokers, a minority of concentrations are extremely high, and these can have a disproportionate impact on results. We therefore used $\log(\text{cotinine} +1)$ as our primary outcome, but also report results using untransformed values (i.e. cotinine concentration) online supplemental appendix A.1. The subset of respondents to the HSE receiving a nurse visit is asked to provide a saliva sample, which is then analysed for cotinine concentration. For 2007, 4058 valid cotinine measurements were available.[i] Current smoking status (smoker/non-smoker) was based on self-report.[ii]

### Socioeconomic deprivation
Our primary measure of socioeconomic deprivation was the 2007 Index of Multiple Deprivation (IMD) quintile.[25] This is an area-based measure linked to postcode, for which we use the quintiles or group into 'deprived' (the lowest two quintiles) and 'non-deprived' (highest three quintiles). IMD was selected over alternative SES variables such as household income and occupation, as it had fewer missing data, it demonstrated the largest differential conditional effect and was more stable during the period between the main survey and nurse follow-up.

### Analysis
#### Regression discontinuity design
We used a sharp regression discontinuity (RD) design to estimate the causal effects of the intervention. The

---

[i]In our main analysis 3718 responses were used. Of these, 309 were excluded due to the bandwidth of the regression discontinuity design and a further 31 were excluded because observations for control variables were missing.

[ii]In the pre-intervention sample, 95.7% of self-reported smokers and 4.3% of non-smokers had cotinine concentrations>12 ng/mL. Reporting error on smoking status, therefore, appears low.

design uses 1 July 2007, the date of implementation of the national smoking ban, as a threshold dividing the sample into 'control' (interviewed before the ban) and 'treated' (interviewed after the ban): with a bandwidth of ±190 days. Doubly robust methods were used, with inverse probability of treatment weights in locally weighted regressions with triangular kernels, to control for potential covariate imbalances at the time of the ban. More details are provided in online supplemental appendix A.2.

The model takes the general form:

$$y = \beta_0 + D\beta_1 + Z\beta_2 + (\boldsymbol{D} \times \boldsymbol{Z})\beta_3 + u \ for \ |\boldsymbol{Z}| \leq h \quad (1)$$

where y is the outcome variable (log cotinine), D is the treatment dummy and Z is the forcing variable, time from ban, which takes the value zero on the day of the policy change, it is positive if the observation is after the ban and negative before. The error term is u, which is assumed to be independent and identically distributed with an expected value of zero. For the coefficients, $\beta_0$ represents the average outcome for the control group at the point of discontinuity where $Z \rightarrow 0$ (date of the ban), $\beta_1$ is the treatment effect at the discontinuity, $\beta_2$ is the time trend before the ban and $\beta_3$ indicates the change in the trend after the ban.

## Treatment effects

We used the RD approach to estimate the potential outcomes individuals would have had with and without treatment at the threshold.[26] The overall impact of the smoking ban on tobacco exposure was measured as an average treatment effect:

$$ATE = \beta_1 = E\left[\boldsymbol{Y}(1) - \boldsymbol{Y}(0) | \boldsymbol{Z} = 0\right]$$

Where Y(0) is the *potential* outcome an individual would have had without treatment and Y(1) is the *potential* outcome with treatment. We then evaluated the distributional impacts of the smoking ban using a series of approaches: (1) conditional average treatment effects (CATEs) (2) quantile treatment effects (QTEs); (3) conditional QTEs (CQTEs) and (4) inequality treatment effects (ITEs).(2)

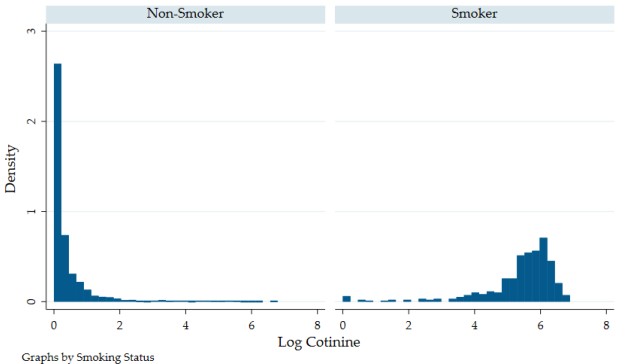

**Figure 1** Distribution of log cotinine: non-smokers and smokers.

CATEs estimate the average effect of the treatment (the smoking ban) on y (log cotinine) conditional on x (eg, whether the effect of the smoking ban varied with socioeconomic deprivation):

$$CATE\left(\tilde{\boldsymbol{x}}\right) = E\left[\boldsymbol{Y}(1) - \boldsymbol{Y}(0) | \tilde{\boldsymbol{X}} = \tilde{\boldsymbol{x}}, \boldsymbol{Z} = 0\right] \quad (3)$$

QTEs estimate the effect of the treatment on *y* at different quantiles (τ) of *y*:

$$QTE(\tau) = Q^\tau_{Y(1)|Z=0} - Q^\tau_{Y(0)|z=0} \quad (4)$$

where $Q^\tau_{Y(.)|Z=0}$ is the quantile function, which returns the cotinine level Y(.) at the threshold for each quantile, $\tau \in (0,1)$ (conditional rank of the outcome). The differences between these quantile functions give the treatment effect at each quantile. These effects are estimated directly, using quantile regressions,[27] within the RD framework.[28] This allows us to determine how the ban changed the distribution of tobacco exposure, revealing whether levels fell more at the top or bottom of the distribution.

CQTEs combine QTEs and CATEs, estimating the effect of the treatment at the quantile of y (log cotinine), conditional on x (eg, socioeconomic deprivation):

$$CQTE(\tau, \tilde{x}) = Q^\tau_{Y(1)|\tilde{X}=\tilde{x}, \ z=0} = Q^\tau_{Y(0)|\tilde{X}=\tilde{x}, \ z=0} \quad (5)$$

Finally, ITEs[29] identify the difference in inequality indices between the treated and control:

$$ITE = W\left(Y(1)|\boldsymbol{Z}=0\right) - W\left(Y(0)|\boldsymbol{Z}=0\right) \quad (6)$$

where W(.) denotes an inequality index, in this case, the Concentration Index,[30] a *relative* measure of bivariate inequality:

$$W\left(y, \tilde{x}\right) = \frac{1}{N} \sum_{i=1}^{N} \left(\left(2R\left(\tilde{x}_i\right) - 1\right)\frac{y_i}{\bar{y}}\right) \quad (7)$$

where $\left(2R\left(\tilde{x}_i\right) - 1\right)$ is the fractional rank of $\tilde{x}_i$ and $\bar{y} = \frac{1}{N}\sum_i^N (y_i)$ is the mean outcome. The concentration index indicates how concentrated health-related outcomes are among those of a higher/lower SES. Where there is no socioeconomic-related inequality, the concentration index is zero, with values further from zero indicating greater inequality. We used methods developed by Heckley *et al*[31] to estimate ITEs within the RD framework using recentered influence function regressions.

## RESULTS

### Baseline characteristics

Figure 1 shows the distribution of transformed cotinine values (log(cotinine+1)) for non-smokers and smokers in the control group (preintervention). For non-smokers, the distribution was heavily right skewed (31.4% of the sample had zero values) with a mean of 0.43 (actual value of 4.95 ng/mL). For smokers, the distribution was left skewed with a mean of 5.37 (303.24 ng/mL).

Baseline characteristics, by quintile of cotinine, are shown in online supplemental appendix A.3. Higher cotinine levels were associated with younger age, being

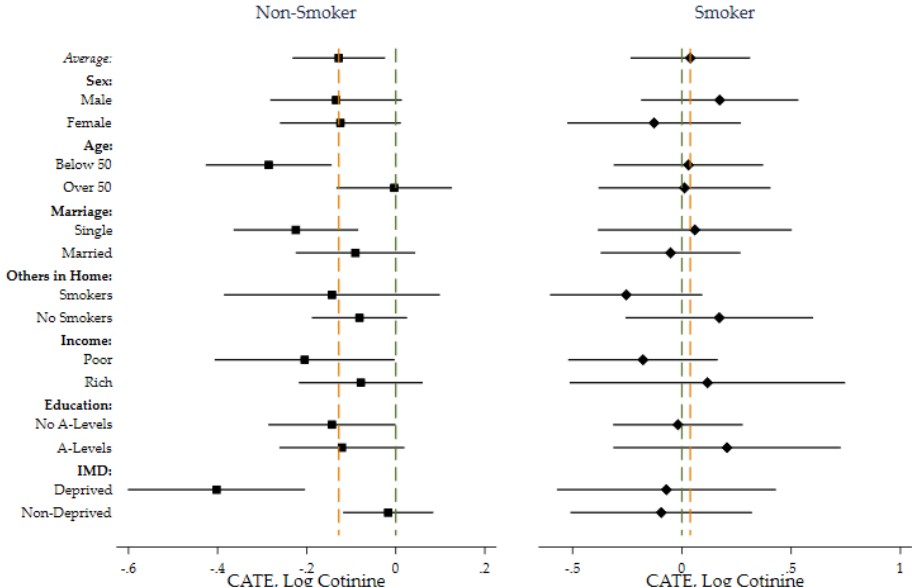

**Figure 2** Forest plot of conditional average treatment effects of log cotinine: non-smokers and smokers key: bars denote 95% CIs. CATE, conditional average treatment effect; IMD, Index of Multiple Deprivation.

unmarried, living in an urban environment, lower SES, poorer self-assessed health and higher levels of alcohol consumption, in addition to smoking status and tobacco consumption.

### Average treatment effects

At the time of the ban, the expected baseline level of log cotinine (without treatment) was 0.40 for non-smokers and 5.34 for smokers. There was a large and significant reduction in cotinine concentrations for non-smokers (−0.128, p=0.043), but no significant effect of the ban for smokers (0.036, p=0.817). Regression outputs are shown in online supplemental appendix A.4. These results are robust to a set of sensitivity analyses, see online supplemental appendix A.5.

These effects are shown at the top of figure 2, which shows forest plots of average effects of the smoking ban conditional on sex, age, marital status, smoking status of others in the household, equalised household income, highest qualification and IMD (CATEs). For smokers, there were no significant effect conditional on any variable. For non-smokers, there were both significant effects and significant differences between subgroups, most notably larger reductions in cotinine levels for younger, single, income poor or IMD-deprived respondents.

The differences in CATEs for non-smokers are partially explained by different baseline predictions across subgroups. For example, the predicted average level of log cotinine at the time of the intervention was 0.65 for deprived and 0.30 for non-deprived. After the ban, these levels fell to 0.23 and 0.27, respectively (not significantly different), see online supplemental appendix A.6 for more detail.

### Distributional analysis

Figure 3 shows QTEs, that is, the effect of the smoking ban on the level of cotinine at different parts of the distribution (quantiles of cotinine). For non-smokers, there was no effect on cotinine levels at the lower quantiles (which were already at 0) but significant, and increasing, reductions appeared at higher quantiles of cotinine. For smokers, there were no significant effects at any quantile.

Figure 4 shows QTEs conditional on IMD deprivation. Significant reductions in cotinine levels are apparent for deprived non-smokers at higher levels of exposure. For non-deprived non-smokers, there are significant small effects at the mid-range of exposure (going from very little to zero exposure) but not at higher levels. For both groups of smokers, there was no significant reduction at any quantile of cotinine.

### Changes in inequality

Table 1 shows the causal effect of the ban on bivariate inequality (the concentration index), that is, how the concentration of log cotinine levels among the socioeconomically deprived (IMD quintile) changes.

The negative constants in models (1) and (2) show that higher levels of cotinine were concentrated among more socioeconomically deprived respondents at baseline for the whole sample (indicating higher levels of smoking) and non-smokers (indicating higher levels of second-hand exposure). However, the constant is close to zero for smokers, therefore conditional on being a smoker, deprivation has limited impact on cotinine levels.

The treatment effect shows that the ban caused a significant increase in the concentration index for non-smokers (ie, towards zero), reducing the concentration

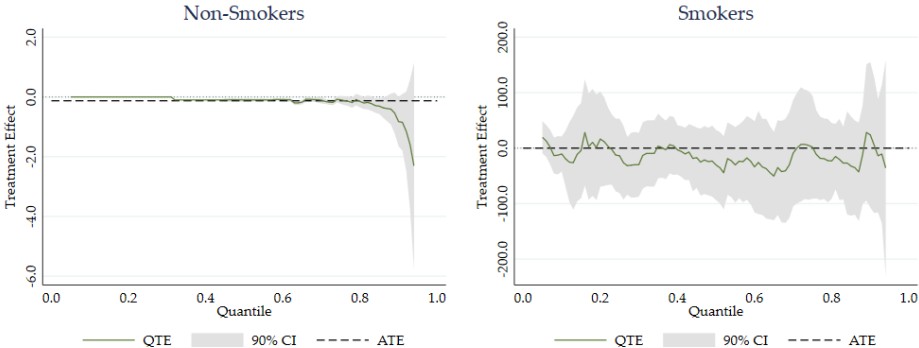

**Figure 3** QTEs: log cotinine QTEs for smokers and non-smokers key: solid line denotes QTE; dotted line denotes ATE; grey area denotes 90% CI. ATE, average treatment effect; QTE, quantile treatment effect.

of cotinine levels among deprived non-smokers. No significant effects were found for the whole sample or for smokers. These results indicate that the ban caused a reduction in deprivation-related inequality in cotinine levels among non-smokers but had no significant effect for smokers.

## DISCUSSION

Given the socioeconomic gradient of tobacco smoke exposure, public smoking bans could potentially benefit deprived groups more. Conversely, poorer individuals may respond less well to such policies due to lack of opportunities and resources, contextual factors[32] and attitudes towards antismoking policies.[33] We found that the smoking ban in England significantly reduced exposure for non-smokers with greater reductions for

socioeconomically deprived groups with higher baseline levels of exposure. However, the ban had no significant impact on exposure levels, or on inequalities in exposure, for smokers.

## Strengths and limitations of the study

We used a doubly robust RD design to derive causal estimates of the differential impacts of a national smoking ban across the entire distribution of exposure levels, for both smokers and non-smokers. We also used a rich, individual-level data set with an objective measure of tobacco smoke exposure to avoid social desirability bias in reported variables and explored differential effects of the ban both between groups using CATEs and across the entire distribution of cotinine levels using quantile treatment effects. We unified these approaches to examine conditional distributional effects and estimate the causal

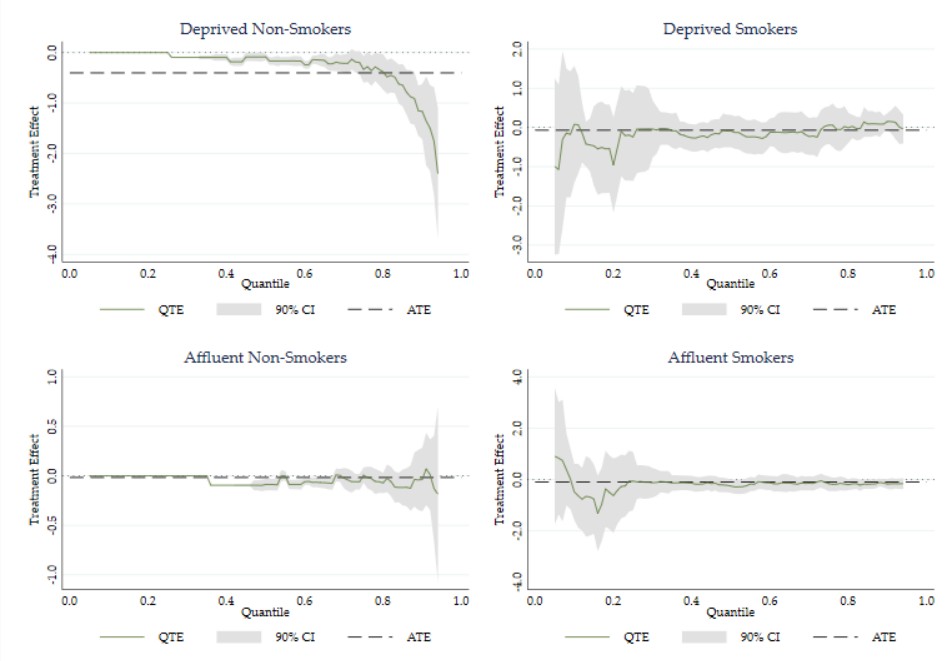

**Figure 4** Conditional QTEs: log cotinine key: solid line denotes QTE; dotted line denotes ATE; grey area denotes 90% CI. ATE, average treatment effect; QTE, quantile treatment effect.

**Table 1** Log cotinine: concentration index

|  | (1) | (2) | (3) |
|---|---|---|---|
|  | Sample | Non-smokers | Smokers |
|  | Coefficient/SE | Coefficient/SE | Coefficient/SE |
| Treatment | 0.0265 | 0.2097** | −0.0285 |
|  | (0.0693) | (0.0910) | (0.0182) |
| Time | 0.0005 | −0.0015* | 0.0003** |
|  | (0.0005) | (0.0008) | (0.0001) |
| Treatment × Time | −0.0009 | 0.0007 | −0.0003 |
|  | (0.0008) | (0.0011) | (0.0002) |
| Constant | −0.1501*** | −0.2329*** | 0.0119 |
|  | (0.0394) | (0.0559) | (0.0127) |
| N | 3718 | 2952 | 766 |
| $R^2$ | 0.0009 | 0.0029 | 0.0076 |

*p<0.10, **p<0.05, *** p<0.01.

impact of the ban on socioeconomic-related inequality in cotinine.

The study had several limitations. First, the HSE is a repeated cross-sectional survey and we were, therefore, unable to observe individual's over time; rather, we compared distributions of exposure for different individuals surveyed before and after the ban. Second, we relied on self-report to determine smoking status at the time of the survey. The distributions of cotinine levels for smokers and non-smokers suggest that self-report was accurate for the majority of subjects, but there may have been some misclassification. Third, social deprivation was measured by area deprivation rather than by individual social status and was arbitrarily categorised in quintiles, our analysis of socioeconomic deprivation was, therefore, less fine grained than our analysis of cotinine levels. Finally, we have only measured the short-term effects of the ban, and these may change over the longer term.

### Findings

Our results suggest that the smoking ban in England did little to reduce overall risk or socioeconomic inequalities in smoking-related diseases for the smoking population, but it did, as intended, reduce both overall risk and inequalities in risk for non-smokers, with greater reductions in environmental tobacco smoke exposure for socioeconomically deprived non-smokers with higher baseline levels of exposure. These findings align with systematic reviews showing that compulsory smoking restrictions are generally more equitable than voluntary policies[34] and Semple et al[21]'s findings of greater reductions in environmental smoke in pubs in more deprived areas following the UK ban. A previous study of cotinine levels in non-smokers by Sims et al[35] similarly reported significant reductions in average exposure following the ban but, in contrast to our results, found no significant reductions in the lowest SES households and a significant

reduction for higher SES households (a 37% reduction in the geometric mean of cotinine), implying an increase in bivariate inequality, whereas our study used an RD approach. Sims et al[35] modelled impacts based on preintervention trends that indicated exposure was already falling prior to 2007. However, these models lacked data for the 3 years immediately prior to 2007 and may have underestimated additional reductions resulting from the ban. Sims et al[35] also measured SES using social class of the household head, resulting in a relatively small sample of low status non-smokers; our different results might, therefore, be explained by reduced levels of exposure being achieved in more deprived areas generally, but not in the very lowest status households. This is consistent with our previous study, which found significant reductions in self-reported exposure for non-smokers, which increased with social deprivation, with the exception of individuals with extremely low SES, for whom there was no significant impact.[20]

A major concern at the time of the ban was that smokers would substitute smoking in private areas for smoking in public places, and that this could unintentionally increase risk for vulnerable groups, as happened in the US following similar bans.[36] Our results suggest that such substitution activity did not occur and that there was a net reduction in exposure for adult non-smokers; this is supported by findings of reductions in smoking prevalence inside cars (from 32% to 26%) and the home (from 65% to 55%) reported by smokers following the ban.[17]

### CONCLUSIONS

The 2007 smoking ban in England reduced environmental tobacco smoke exposure for non-smokers, with the largest benefits for more deprived non-smokers with high levels of exposure. This group was most at risk from the negative externalities of smoking, and the ban was, therefore, not only an effective but an equitable intervention. Nevertheless, there might still have been a negative equity impact of the ban across the whole population due to an increased gap in exposure between smokers (who are more likely to be deprived) and non-smokers (who are more likely to be affluent). However, our results show that gaps in overall levels of tobacco smoke exposure narrowed between affluent and deprived non-smokers after the ban and bivariate inequality across the whole population did not increase.

**Acknowledgements** We would like to thank Richard Cookson, Owen O'Donnell, Elena Ratschen, Kamran Siddiqi and Tom Van Ourti for their comments and enlightening debate.

**Contributors** Authors contributed equally to the paper. JL extracted the data, performed initial analysis and wrote the first draft. MR developed the methods, performed the final analysis and wrote up the methods, results and appendices. TD wrote the abstract, introduction and discussion. All authors critically edited the manuscript.

**Funding** The study was funded by the Wellcome Trust (205427/Z/16/Z). All errors remain our own.

**Competing interests** None declared.

**Patient consent for publication** Not required.

**Ethics approval** This paper performs secondary analysis of publicly available data, therefore, ethical approval was not needed.

**Provenance and peer review** Not commissioned; externally peer reviewed.

**Data availability statement** Data may be obtained from a third party and are not publicly available. Data are from the Health Survey for England, 2007 Wave. This is accessible through the UK Data Service, persistent identifier: 10.5255/UKDA-SN-6112-1. This data is safeguarded, and accessible through a end user license.

**ORCID iD**

Matthew Robson http://orcid.org/0000-0003-4558-7637

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
