## [Reviewer comments · BMJ Open]

ARTICLE DETAILS

TITLE (PROVISIONAL)	Estimating the Equity Impacts of the Smoking Ban in England on Cotinine Levels: A Regression Discontinuity Design
AUTHORS	Robson, Matthew; Lord, Joseph; Doran, Tim

VERSION 1 – REVIEW

REVIEWER	Frazer, Kathleen University College Dublin, Nmhs
REVIEW RETURNED	15-Mar-2021

GENERAL COMMENTS	This is an important paper presenting additional evidence of the impact of English smoking bans on reducing secondhand smoke exposure for non smokers. Evidence presents statistical robust analyses and acknowledgement of limitation of cross-sectional data sets. The need for additional research and evidence on marginalised groups was acknowledged in a 2016 Cochrane Review of the evidence supporting legislative smoking bans. This study provides more data.
--

REVIEWER	TCHICAYA, Anastase Luxembourg Institute of Socio-Economic Research (LISER), Living Conditions
REVIEW RETURNED	05-Apr-2021

GENERAL COMMENTS	This manuscript deals with an interesting public health policy topic regarding smoking in England. The authors aim to estimate the equity impacts of the 2007 smoking ban in England for both smokers and non-smokers. However, many aspects need to be addressed. 1- The title of this manuscript should be more informative. This study is a cross-sectional study that assesses the short-term impact of the national smoking ban on July 1, 2007.2. There is a difference between the number of participants listed in the abstract (3718, including 766 smokers) and the number of valid cotinine readings (4058). The authors could explain such a difference and could rewrite the footnote 1.3. When was the health survey for England in 2007?4. In the subsection "Average Treatment Effects", the correct figure number is Figure 9 in Appendix A.6 and not Figure 6.5. The discussion does not sufficiently highlight the results of this study compared to others. For example, what would be the differences in the magnitude of equity's impacts compared to other
---

	studies? There is much emphasis on models and not enough on results. 6. Although the authors used an objective measure of tobacco exposure, namely the level of salivary cotinine concentration, the short-term assessment of the impact of banning smoking in public spaces on smokers through this measure seems problematic. Some smokers have likely reduced their level of tobacco consumption or smoking quit. 7. Furthermore, bibliographic references' presentation does not seem to comply with the journal BMJ Open guidelines.
--	--

VERSION 1 – AUTHOR RESPONSE

Reviewer: 1

Dr. Kathleen Frazer, University College Dublin

-Thank you very much for this comment. We are glad to be providing more robust analysis, particularly on marginalised groups.

Reviewer: 2

Dr. Anastase TCHICAYA, Luxembourg Institute of Socio-Economic Research (LISER)

1- The title of this manuscript should be more informative. This study is a cross-sectional study that assesses the short-term impact of the national smoking ban on July 1, 2007.

-We have changed the title to: “Estimating the Equity Impacts of the Smoking Ban in England on Cotinine Levels: A Regression Discontinuity Design”, to make it more informative and to conform to the journal style, and the comment from the editor and yourself. We hope that this fits with your suggestion. We have used “Regression Discontinuity Design”, rather than “cross-sectional study” as this more accurately represents the conducted analyses, and implicit in that is that RDDs investigate short-term impacts.

2. There is a difference between the number of participants listed in the abstract (3718, including 766 smokers) and the number of valid cotinine readings (4058). The authors could explain such a difference and could rewrite the footnote 1.

Thank you for pointing this out. We have now included a new footnote 1, which gives the reason for this discrepancy:

“ In our main analysis 3,718 responses were used. Of these, 309 were excluded due to the bandwidth of the regression discontinuity design and a further 31 were excluded because observations for control variables were missing.”

The main reason for this is the bandwidth used for the RDD. This excludes observations outside of the 190+- day cut-off. The choice of this cut-off, as we show in appendix A.5, does not make much of a difference to our results.

The additional 31 were excluded because observations for the control variables used to estimate the propensity weights were missing.

We have also added more information to the previous Footnote 1, now Footnote 2, to qualify the reason behind statement made.

3. When was the health survey for England in 2007?

The HSE does not occur on a fixed day, like a Census, rather surveyors visit households throughout the year – this feature of the survey is crucial for the RDD study design, which relies on responses to the survey occurring both before and after the date of the smoking ban's implementation. The 2007 Wave of the Health Survey for England started in January 2007 and sampling ran each month until the end of the year. However, as with every year's survey, there was some spill over into the next year, particularly with the Nurse's surveys, which can take place up to 3 months after the initial survey. So, we see 7 respondents in April 2008 (the latest month), for example. However, as we use the threshold for the RDD, this restricts the sample to between January 2007 and January 2008.

We have included these dates in the description of the data to make this clearer.

4. In the subsection "Average Treatment Effects", the correct figure number is Figure 9 in Appendix A.6 and not Figure 6.

Thank you for drawing our attention to this error, we have now corrected it.

5. The discussion does not sufficiently highlight the results of this study compared to others. For example, what would be the differences in the magnitude of equity's impacts compared to other studies? There is much emphasis on models and not enough on results.

The lack of comparison of our results with those of other papers is primarily due to the lack of comparable results in previous studies. No other papers that we know of have estimated Quantile Treatment Effects, Conditional Quantile Treatment Effects or Inequality Treatment effects of the ban (or other bans) – indeed, this was one of the motivations for writing the paper. Although we dichotomise our results in a similar way to previous studies in Figure 2 (e.g. by classifying households as 'high' or 'low' SES), our intention was to go further and quantify the effects of the ban across the whole distribution of SES. Doing this allows us to provide a much more nuanced analysis than has been previously attempted. This means that while we can make statements relating to the equity impacts of the ban, we cannot directly compare our findings to those of previous studies.

However, we recognize that it is also important to engage with previous research and to explain any apparent discrepancies. The only previous analysis which goes part way to estimating equity impacts on non-smokers is the Sims et al paper. They performed subgroup analysis, using the social class (occupation) of the household head, to estimate effects for low SES and high SES individuals. This is similar to our CATE analysis (Figure 2). They reported a reduction in cotinine levels for high SES households but not for low SES households, implying an increase in inequality, although they not discuss

this or quantify changes in inequality. We have now added additional material to the discussion to clarify what their results imply for equity impacts:

“A previous study of cotinine levels in non-smokers by Sims et al. similarly reported significant reductions in average exposure following the ban but, in contrast to our results, found no significant reductions in the lowest SES households and a significant reduction for higher SES households (a 37% reduction in the geometric mean of cotinine), implying an increase in bivariate inequality [35]. Whereas our study used a regression discontinuity approach, Sims et al. modelled impacts based on pre-intervention trends that indicated exposure was already falling prior to 2007. However, these models lacked data for the three years immediately prior to 2007, and may have underestimated additional reductions resulting from the ban. Sims et al. also measured SES using social class of the household head, resulting in a relatively small sample of low status non-smokers; our different results might therefore be explained by reduced levels of exposure being achieved in more deprived areas generally, but not in the very lowest status households.”

For the effects on smokers, existing studies similarly show that there is no effect of the ban on smoking prevalence or intensity (see next response). Partially due to this, these studies do not estimate equity impacts. As there are no (short run) impacts on smokers, there are no equity impacts.

Ours is also the only study to look at the effect of the ban on cotinine for both smokers and non-smokers. Therefore, it is the only paper to identify the effect on bivariate inequality for the whole population (we find a positive but non-significant effect).

For the above reasons, we placed a lot of emphasis on our models, and we hope that others will be able to use similar methods in future papers to allow the estimation of equity impacts. As we note in the paper, it is vitally important to look at the full distribution of treatment effects, we’re finding the largest impacts on the poor non-smokers at the top of the distribution (i.e. those with the highest exposure).

6. Although the authors used an objective measure of tobacco exposure, namely the level of salivary cotinine concentration, the short-term assessment of the impact of banning smoking in public spaces on smokers through this measure seems problematic. Some smokers have likely reduced their level of tobacco consumption or smoking quit.

We agree that one might expect an immediate reduction in tobacco consumption and smoking rates in response to the smoking ban, but we and others have found that this did not occur. The results in our paper show that while there was an effect of the ban on the cotinine levels of non-smokers, there was no such effect on smokers. Neither the ATEs, CATEs, QTEs or ITEs show any effect of the ban on reducing cotinine levels of smokers. This finding appears to be very robust and is corroborated by previous research into smoking habits in the aftermath of the ban (e.g. Fowkes, F, Marlene, C, Stewart, F, et al. “Scottish smoke-free legislation and trends in smoking cessation”. *Addiction* 103.11 (2008), 1888–1895. Lee, J, Glantz, S, and Millett, C. “Effect of smoke-free legislation on adult smoking behaviour in England in the 18 months following implementation”. *PLoS One* 6.6 (2011), e20933.) In a previous study on self-reported measures of smoking for smokers and non-smokers, we also found no effects of the ban on smoking prevalence nor on the number of cigarettes smoked.

(<https://www.york.ac.uk/media/economics/documents/hedg/workingpapers/1920.pdf>).

However, what our current paper shows is that while tobacco consumption did not change, smoking behaviour did. Smokers smoked significantly less in pubs and at work (Lee et al made a similar observation) and their propensity to smoke near vulnerable groups was significantly reduced.

The above results support our conclusion that, in the short run, levels of tobacco consumption by smokers did not change but smokers changed where they smoked and whom they smoked around, leading to the reduction in second hand smoke exposure we observed.

In this study, we did not investigate long run effects. We therefore cannot generate a causal estimate of the smoking ban on long run smoking behaviour. It is possible that the ban helped to reduce smoking prevalence in the long run, but we cannot identify that causally with our data or experimental design.

We have tried to bring out the above issues more clearly in the revised paper, by making the points clearer to the reader in several places. For example, we have rephrased the introduction to be more explicit on the findings of the existing literature, and have included the reference to our previous paper there.

7. Furthermore, bibliographic references' presentation does not seem to comply with the journal BMJ Open guidelines.

We have now changed all the references to comply with the reference style of BMJ Open.

VERSION 2 – REVIEW

REVIEWER	TCHICAYA, Anastase Luxembourg Institute of Socio-Economic Research (LISER), Living Conditions
REVIEW RETURNED	11-Aug-2021
GENERAL COMMENTS	Thanks to the authors. After the modifications made by the authors in this manuscript, it is potentially acceptable for publication in BMJ OPEN